## Research Article

mHealth; early childhood development; global mental health; community-based intervention; user-centered design

**Corresponding author:**
A. Desrosiers;
Email: alethea_desrosiers@brown.edu

# Applying user-centered design to enhance the usability and acceptability of an mHealth supervision tool for community health workers delivering an evidence-based intervention in rural Sierra Leone

Cara M. Antonaccio[1] [iD], Justin Preston[1], Chokdee Rutirasiri[2],
Sunand Bhattacharya[2], Musu Moigua[3], Mahmoud Feika[3] and Alethea Desrosiers[1] [iD]

[1]Alpert Medical School, Brown University, Providence, RI, USA; [2]School of Social Work, Boston College, Chestnut Hill, MA, USA and [3]Caritas Freetown, Freetown, Sierra Leone

## Abstract

Mobile health (mHealth) platforms have the potential to increase access to evidence-based interventions in low-resource settings. This study applied a user-centered design (UCD) approach to develop and evaluate an mHealth supervision tool for community health workers (CHWs) delivering an early childhood development intervention in rural Sierra Leone. We engaged CHWs (N=8) and supervisors (N=4) in focus group discussions, user testing sessions and exit interviews to gather feedback on the mHealth supervision tool's usability and acceptability. Mixed methods findings indicated that the tool was generally well-received and perceived as easy to use, but there were also challenges related to connectivity, phone charging and the need for more comprehensive training and support. Overall, this study suggests that a UCD approach can promote the usability of mHealth tools to support CHWs in delivering evidence-based interventions in low-resource settings, highlighting the importance of addressing contextual challenges and providing adequate training and support to ensure the effectiveness and sustainability of such tools.

## Impact statement

This study illustrates the user-centered design process and evaluation of a mobile health (mHealth) supervision tool to enhance the delivery of an evidence-based early childhood development intervention in rural Sierra Leone. By actively involving community health workers and supervisors in the design and development of the mHealth supervision tool, our goal was to tailor the mHealth tool to their specific needs and challenges. We found that the mHealth tool met supervision needs and empowered users with new technological skills. The user-centered design process in this study has the potential to be replicated and scaled in similar low-resource settings. By bridging the gap between technology, workforce capacity and community-based care, this study contributes to the growing body of evidence supporting the use of mHealth strategies to address key global health inequities.

## Introduction

Intergenerational cycles of trauma and violence, coupled with limited societal infrastructure and economic opportunities, pose significant risks for millions of children and families in conflict-affected communities. In Sierra Leone, two decades after an 11-year civil war and nearly a decade after the 2014–2016 Ebola outbreak, child mortality rates remain among the highest globally, and child physical abuse and maltreatment are prevalent (UNICEF, 2024). Resource constraints have limited the government's ability to address these challenges. However, several programs, such as the Interaction Competencies for Teachers (Masath et al., 2020) and the Family Strengthening Intervention for Early Childhood Development plus Violence Prevention (FSI ECD+VP/Sugira Muryango; Betancourt et al., 2020a), have effectively increased access to evidence-based child development interventions by training nonspecialists to deliver the intervention and by using alternative delivery approaches (i.e., home delivery) in resource limited settings within sub-Saharan Africa.

While evidence-based interventions (EBIs) like the FSI-ECD+VP demonstrate promise for supporting children and families affected by adversity, access and maintaining fidelity to intervention delivery remain key challenges in settings like the rural regions of Sierra Leone,

largely due to implementation barriers (i.e., limited health care workforce, transportation difficulties, poor infrastructure; Lyon and Koerner, 2016). Innovative approaches are needed to overcome these barriers to accessing EBIs that promote positive parenting practices and early childhood development outcomes for families with young children. The use of trained nonspecialists to deliver EBIs, combined with mobile health (mHealth) strategies, may help increase the reach and efficiency of service delivery by optimizing resource utilization and improving outcomes (Bunn et al., 2021; Desrosiers et al., 2021; Mudiyanselage et al., 2024; Winters et al., 2019). For example, mHealth strategies can support the ongoing supervision and quality improvement of CHWs providing in-home services (Triplett et al., 2023). In rural areas of Sierra Leone, where infrastructure challenges are common (i.e., transportation costs, poor roads, low internet connectivity,frequent power outages), mHealth tools can address these challenges during the design and development process by incorporating features like offline functionality, access to cloud storage and using battery-powered tablets. Using mHealth tools with offline functionality can reduce the need to travel for supervision, thus reducing the burden of costs (i.e., fuel) and time related to transportation for in-person meetings or Wi-Fi access.

Integrating user-centered design (UCD) principles in the development and tailoring of mHealth strategies is a critical advancement for effective implementation (Ettinger et al., 2016; Poulsen et al., 2023; Stephan et al., 2017). This iterative process centers the perspectives, knowledge and needs of people with lived experience, fostering the co-creation of accessible, feasible and sustainable interventions (Lyon and Koerner, 2016). This approach has been used previously to design and develop a digital tool for training nonspecialists to deliver an evidence-based psychological treatment for depression in primary care in India (Khan et al., 2020).

Building on the potential of mHealth interventions and UCD, the current study applied UCD processes to develop an mHealth-based supervision tool to support delivery of a culturally adapted version of the FSI-ECD+VP in Sierra Leone. The FSI-ECD+VP has demontrated effectiveness in promoting early child development, caregiver mental health and positive parenting practices among families living in extreme poverty in Rwanda (Barnhart et al., 2020; Betancourt et al., 2020a; Jensen et al., 2021). The FSI-ECD+VP consists of 12 modules focused on topics such as the importance of early stimulation for child development, positive parenting practices, conflict resolution, stress management, and father engagement. The FSI-ECD+VP has also shown preliminary benefits for promoting caregiver mental health and preventing household violence among vulnerable families with young children in Sierra Leone (Desrosiers et al., 2024). In this study, we used the analyze, design, develop, implement and evaluate framework (ADDIE; Dick, 1996), a five-phase UCD process, to iteratively design and implement mHealth supervision and fidelity monitoring tools for CHWs and supervisors. By centering the needs and preferences of CHWs and supervisors throughout the development process, we aimed to create a supervision tool that was not only user-friendly and acceptable but also responsive to the practical demands of the context.

## Methods

### Sampling and recruitment

The study protocol was approved by the Boston College Internal Review Board (Protocol #21.006.01) and the Sierra Leone Ethics and Scientific Review Committee. All participants provided oral informed consent before participating in the study. We recruited CHWs (N = 8; four males, four females) from two Peripheral Health Units (PHUs) in rural areas of the Makeni region in Sierra Leone, along with their direct supervisors (N=4; two males, two females). CHWs were eligible if they were 18 years or older, able to commit to attending three 90-min sessions and currently employed in delivering maternal and child health services within the Makeni region. Supervisors were required to be 18 years or older and actively working as supervisors of CHWs providing services in the same region. The PHU Focal Person recommended CHWs and supervisors who were in good standing and expressed interest in the project. The study Project Coordinator contacted potential participants in the order in which referrals were made by the PHU Focal Person. Those who were eligible and provided informed consent were enrolled in the study until the target sample size was reached.

CHWs and supervisors completed a three-week training on the FSI-ECD+VP, which included a combination of didactic instruction on intervention content, role plays and group discussions. Focusing on a specific intervention may have limited the generalizability of our findings to other interventions; however, this focus also allowed us to develop a more tailored and usable mHealth supervision tool, which could support future scale-out of the FSI-ECD+VP and/or be adapted for use with other family home-visiting services to monitor delivery quality and improve feedback cycles during supervision. In addition to training on the FSI-ECD+ VP, participants also completed a 1-day technology training on the mHealth supervision tool.

### ADDIE process framework

CHWs and supervisors served as experts in the mHealth tool design and development process. Before beginning the UCD process, we defined the end goal as the creation of a mobile app to enhance the delivery quality of the FSI-ECD+VP as well as the supervision process between CHWs and supervisors. We also explored CHWs' technical literacy and familiarity with mobile tools (i.e., tablets, mobile phones) in a brief survey to help inform the UCD process and development of the mHealth supervision tool. The survey asked about CHWs' experience using mobile devices, their ability to use basic features such as texting and browsing the internet, and their comfort level with learning new technologies. We then launched the UCD process. Activities during each phase are described below.

### Analyze

The program manager, a member of the in-country research team, facilitated two hybrid problem analysis focus group discussions (FGDs), blending in-person meetings in Makeni with remote teleconference sessions led by the design team. This approach allowed for direct engagement with participants while leveraging the expertise of the design team. The analysis phase explored the current challenges that CHWs and supervisors encountered in their day-to-day practice and pinpointed specific problem indicators, such as the need for improved documentation, data collection and communication between CHWs and supervisors. The insights from this in-depth problem analysis – including user needs, preferences and contextual factors – directly informed the design and development of the mHealth tool to help ensure its relevance and fit within the local health system. The initial problem analysis FGDs, facilitated by the in-country program manager, focused on gaining a

deep understanding of the current service monitoring and supervision processes from the perspectives of CHWs and supervisors. We explored their experiences, challenges and perceived needs, and thoughts on how the integration of mHealth tools could potentially streamline and enhance these processes. Participants were encouraged to share specific recommendations for resources that would be most beneficial in supporting CHWs during the use of mHealth technology. We created a "mind map" (a visual representation of important factors and processes) based on the qualitative findings from FGDs to illustrate the supervision process and how it relates to CHWs and supervisors' specific needs. The mind map revealed key challenges faced by CHWs and supervisors, such as the need for improved documentation, data collection and communication tools. These findings directly informed the design and development of the mHealth tool.

### Design and develop

Leveraging findings from the problem analysis phase, we developed an initial prototype of the mHealth app that incorporated the contextual findings from the analysis phase. The design team included two faculty members at Boston College with extensive prior experience in UCD processes as well as one postdoctoral fellow who provided support. The primary design team leader (one faculty member) facilitated hybrid teleconference sessions remotely, while the in-country program manager convened the CHWs and supervisors in-person. We then conducted two rounds of iterative user interface/user experience testing (UI/UX) with both CHWs and supervisors via this hybrid format.

User testing sessions were guided by the Think-Aloud Testing protocol method (Charters, 2003), which encouraged participants to articulate their thoughts and actions in real-time as they interacted with the mHealth tool. During the think-aloud testing session, we asked a series of questions to understand CHWs' experiences with the mHealth supervision tool. Think-aloud questions focused on the clarity and ease of mHealth tool navigation, the visual appeal and structure of the tool, the readability of the text and understandability of icons or images and any aspects that were confusing or challenging. We also asked participants for feedback on potential features to add or remove, what features they thought were the strongest, and whether they believed the tool would be helpful for supervision, performance monitoring and useful for other CHWs and supervisors. Each user-testing session was audio-recorded, translated and transcribed. Real-time observations and feedback from user testing sessions directly informed iterative refinements to the prototype to enhance its user-friendliness.

### Implement

Following the design and development phases, the final mHealth supervision tool prototype was implemented by CHWs and supervisors who delivered the FSI-ECD+VP to families with young children in rural areas of the Makeni region. While the mHealth tool prototype was finalized before implementation, we collected feedback through interviews and surveys to inform potential updates and improvements to consider in the future. The phones used in the study were supplied by the research team to the CHWs and supervisors for the duration of the project. CHWs used the mHealth tool to help guide the delivery of session content, track progress with different families and monitor their delivery quality (i.e., fidelity to session content and competency). Sessions were delivered during home visits and recorded by CHWs, with supervisors using the tool to remotely track CHW session delivery progress, assess session delivery quality via digitized fidelity

checklists and identify areas for improvement. Supervisors had access to the same information as CHWs on the mHealth tool, which included the session recordings, tracking tools to monitor CHW progress, and fidelity checklists.

### Evaluate

CHWs and supervisors completed the System Usability Scale (SUS; Brooke, 1996), a validated Likert-style questionnaire measuring perceived usability, before the FSI-ECD+VP was delivered and during the evaluation phase after implementation of all FSI-ECD+-VP sessions had concluded. CHWs and supervisors also completed qualitative exit interviews exploring them Health tool's feasibility, acceptability and usability. All interviews were audio-recorded, translated and transcribed. Data from the evaluation phase provided insights into the specific design elements and app functions that resonated with users, as well as potential areas for improvement in future iterations of the mHealth tool.

## Convergent parallel mixed methods design

We used a convergent parallel mixed methods design (Creswell & Plano Clark, 2018) to evaluate the usability, feasibility and acceptability of the codesigned mHealth supervision tool among CHWs and supervisors. This design leveraged the strengths of both qualitative and quantitative data collection and analysis techniques integrating the findings for a deeper exploration of user experiences and perceptions (qualitative) while also providing a structured assessment of the tool (quantitative). The mixed methods analysis process involved the first and second authors comparing qualitative themes with quantitative usability data to examine how, if at all, the mHealth tool aligned with user needs and preferences.

### Data and procedures

Qualitative data were collected from two problem analysis FGDs with CHWs and supervisors, the think-aloud protocol during user testing, and from qualitative exit interviews with CHWs (N=4) and supervisors (N=2) during the evaluation phase. Quantitative data were collected using the SUS, which was administered before and after the implementation of the mHealth tool, to assess changes in user perceptions on mHealth tool usability.

### Analytical approach

Qualitative and quantitative data were analyzed separately and then integrated in a joint display figure. Paired $t$ tests were used to examine mHealth system usability perceptions pre- and post-implementation. Qualitative themes were compared with quantitative usability perceptions to assess whether the tool's design and implementation aligned with user needs and preferences. This triangulation process helped to promote the comprehensiveness of the findings. The first and second authors (CA, JP) independently coded qualitative transcripts from FGDs and exit interviews with CHWs and supervisors, applying a combination of inductive and deductive coding techniques. Inductive codes emerged organically from the data, while deductive codes were derived from the study's aims and existing literature.

We then used reflexive thematic analysis (RTA; Braun and Clarke, 2019) to systematically identify, analyze and interpret patterns (themes) in the qualitative data. We sought to gain a comprehensive understanding of the challenges faced by CHWs and supervisors in their current roles, their specific needs and preferences regarding the mHealth supervision tool and the potential impact of such tools on their work experiences and service delivery.

The iterative nature of RTA allowed for the refinement of codes and themes throughout the analysis process and facilitated ongoing awareness of the potential influence of subjective biases related to the interpretation of findings. This reflexive approach ensured that the final thematic framework accurately reflected the complexities and nuances of the data.

## Results

The demographic characteristics of the participants are presented in Table 1. The sample (N=12; eight CHWs and four supervisors) was equal in terms of representation from male and female CHWs and supervisors. CHW participants were 36.1 years old, on average (SD=9.1) and of the eight CHWs, one had 1 year of experience in their role, four had between 2 and 5 years of experience and three had between 6 and 10 years. Supervisors ranged in age from 37 to 50 years.

### ADDIE process

Findings from FGDs during the analysis phase indicated that CHWs and supervisors' existing service delivery model primarily relied on in-person interactions, manual note-taking and occasional audio recordings. CHWs and supervisors expressed a desire for tools that could streamline documentation, improve communication and feedback and facilitate data collection and progress tracking. The problem analysis FGD findings and the resulting mind map informed the organization of the components of the mHealth tool's information infrastructure and UI/UX. For example, FGDs identified the need for streamlined documentation and improved communication in the supervision process; this feedback informed the design and development of the tool's digital supervision checklist features. Similarly, the UI/UX was influenced by the need to accommodate varying levels of technological literacy, which necessitated a simple and intuitive interface with clear navigation and prominent icons.

The results of the design and development phases are demonstrated in Figures 1 and 2. Figure 1 demonstrates the information architecture of the mHealth supervision tool, illustrating how the tool's components work together. Figure 2 exhibits the first

**Table 1.** Demographic characteristics of CHWs and supervisors

| Characteristic | | CHWs (N=8) and supervisors (N=4) | |
|---|---|---|---|
| | | Mean | SD |
| Age, mean (years) | | 36.1 | 9.1 |
| | | N | % |
| Age | 18–25 | 2 | 16.7 |
| | 26–35 | 5 | 41.7 |
| | 36–45 | 2 | 16.7 |
| | 45–55 | 3 | 25.0 |
| Gender | Female | 6 | 50.0 |
| | Male | 6 | 50.0 |
| CHWs' years of experience | 0–1 | 1 | 8.4 |
| | 2–5 | 4 | 50.0 |
| | 6–10 | 3 | 37.5 |

prototype of the mHealth supervision tool including the UI, core features, and how the various functionalities are used in the supervision process. CHWs and supervisors received a brief training on the functions and features of the tool. CHWs and supervisors completed a 1-day, in-person technology training on the use of the mHealth tools. The training plan was guided by feedback gathered throughout the design process and included a walkthrough of the key functions and descriptions of each (see Figure 3). The training involved hands-on practice and technical assistance to troubleshoot questions. It sought to guide CHWs and supervisors through the features and functions of the mHealth tool and the procedures for using it (in accordance with the study protocol) during their home-visiting sessions with each family. After the training and before implementation, CHWs generally expressed positive perceptions of the tool's usability. They indicated a willingness to use the system frequently (mean = 4.5 out of 5) and felt confident in their ability to do so (mean = 4.5). The system was perceived as fairly easy to use (mean = 4.14) and not overly complex (mean = 2.38), with well-integrated functions and components (mean = 4.25). However, there was a moderate perceived need for technical assistance (mean = 3.13) and some indication of inconsistency within the system (mean = 2.13).

### Pre- to post-implementation system usability findings

A mixed methods joint display matrix is presented in Table 2 to demonstrate mHealth system usability findings via the integration qualitative and quantitative user perceptions. The triangulation of quantitative usability data with qualitative evidence from each stage of the UCD process revealed that the mHealth supervision tool met many of the needs and preferences of CHWs and supervisors and it helped to improve their ability to deliver the FSI-ECD+VP sessions with quality. Post-intervention usability findings similarly indicated that CHWs generally found the mHealth supervision tool easy to use (mean = 4.13) and felt confident using it (mean = 4.5). CHWs also reported a high likelihood after using the mHealth tool that they would use the system frequently (mean = 4.75) and perceived the system's functions and components as well-integrated (mean = 4.5).

However, CHWs also found the system somewhat complex (mean = 2.88) and difficult to use (mean = 2.75) and expressed a need for technical assistance (mean = 4.75). The results of the paired *t* test suggest that, compared to before implementation, CHWs had mixed perceptions of the tool's usability after using it during FSI-ECD+VP implementation (See Table 2). Small, non-significant increases were observed in the CHWs' desire to use the system (i.e., the mHealth tool) frequently and in their confidence in using the system. However, there were also small, non-significant increases in the perceived complexity and difficulty of using the system. The only statistically significant change in usability was an increase in the perception that they would need help from a technical person to use the system.

### Strengths of the mHealth system
Both the quantitative usability metrics (high scores on ease of use and usefulness) and qualitative feedback indicated that the mHealth tool was generally well-received and perceived as easy to use and helpful in CHWs' work. For example, one CHW stated, "The experience was good…all the equipment was ok." (CHW 4) Another CHW simply stated, "It's easy to use, I use it well." (CHW 2) Quantitative findings on system usability were consistent with qualitative feedback that highlighted the tool's helpfulness in

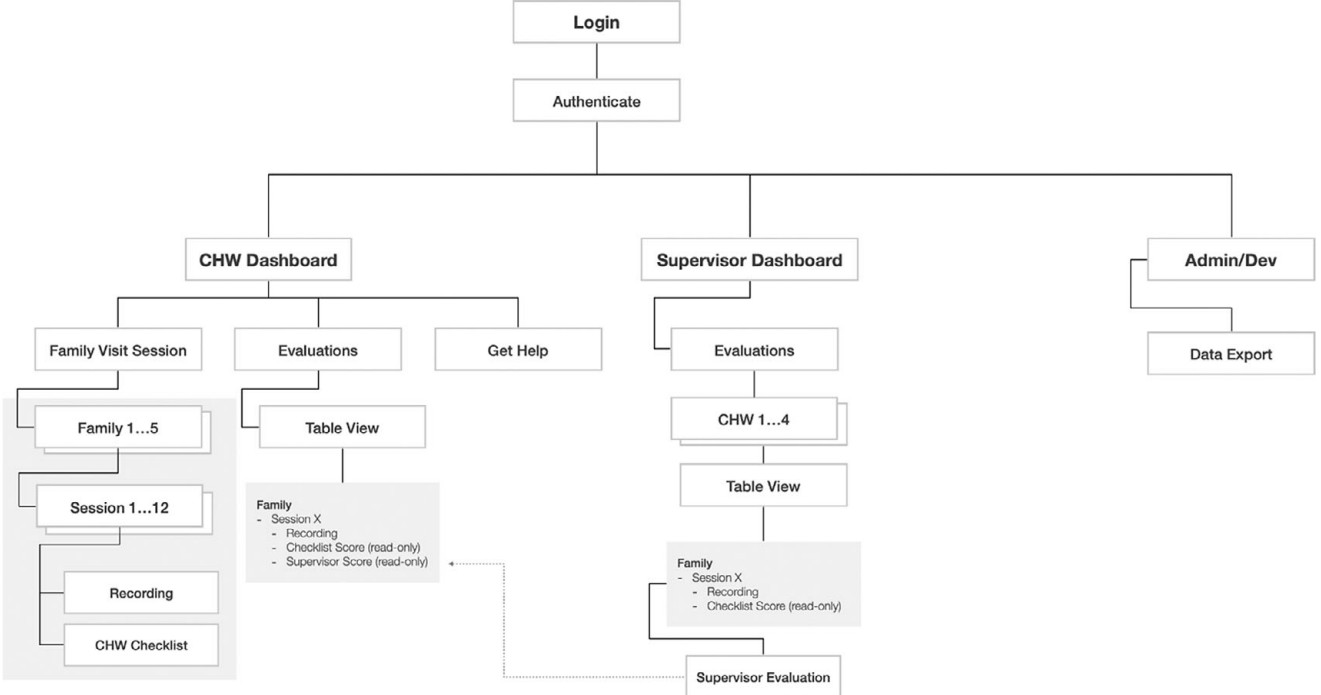

**Figure 1.** mHealth app information architecture (version 1).

providing practical guidance and decision support for home-visiting sessions. One CHW noted, "The tools direct us what to say and during the training we were taught a lot." (CHW 1), while another shared, "It helps me a lot. I can now advise mothers better." (CHW 2) Qualitative evidence suggests that the mHealth tool positively impacted service delivery by enhancing CHWs' ability to conduct home visits, improving communication with supervisors and increasing their knowledge and confidence.

### Areas to improve the mHealth system

In addition to perceived strengths of the tool, CHWs identified ways that the mHealth supervision tool and the way that it was implemented could be improved. For example, quantitative analysis revealed a statistically significant increase in the perceived need for technical support after tool implementation. A slight, though not statistically significant, increase was observed in perceived inconsistency within the system along with a slight decrease in the perception that most people would learn to use the system quickly. This might suggest that the tool's interface and learning curve could be further refined to optimize user experience; and aligns with feedback from a CHW during user testing who stated, "The only concern that I have, the only concern for me, we have not yet learned a lot about it. It's the only concern that I have. They have just introduced it to us, we have not yet learned about it." The qualitative feedback also highlighted challenges with connectivity, inconsistent electricity access for phone charging and requests for more comprehensive training and troubleshooting assistance. One supervisor mentioned, "The only challenge for the cell phone is maybe if we don't have light [electricity]," while a CHW added, "They should add [to] the training." (CHW 3)

### Discussion

This study highlights the potential of mHealth tools to support the delivery and supervision of EBIs in low-resource settings, contributing to a growing body of evidence on the usability of such tools. By centering the needs and preferences of end-users throughout the design process, we developed a tool that was both acceptable and usable for CHWs and supervisors in rural areas of Sierra Leone. The hybrid UCD approach we used can serve as a model for developing and implementing mHealth tools in other resource-constrained settings or for different evidence-based behavioral health interventions, with the primary goal of increasing engagement and adoption of mHealth tools that are user-friendly.

Our findings demonstrate that the mHealth supervision tool was generally well-received by CHWs and supervisors. Both quantitative usability metrics (high scores on ease of use) and qualitative feedback indicated that the mHealth tool was generally well-received and perceived as easy to use and helpful in CHWs' work. Qualitative feedback suggests that the mHealth tool positively impacted service delivery by enhancing CHWs' ability to conduct home visits, improving communication with supervisors and increasing their knowledge and confidence. However, we also identified several areas for improvement. For example, quantitative analysis revealed a statistically significant increase in the perceived need for technical support during tool implementation. Qualitative feedback also highlighted challenges with connectivity, inconsistent electricity access for phone charging and requests for more comprehensive training and troubleshooting assistance.

To address these challenges, future efforts should develop more detailed training materials that cover all aspects of the mHealth tool's functionality. These materials could include digital content from the FSI-ECD+VP manual, "cheat sheets" on key session topics

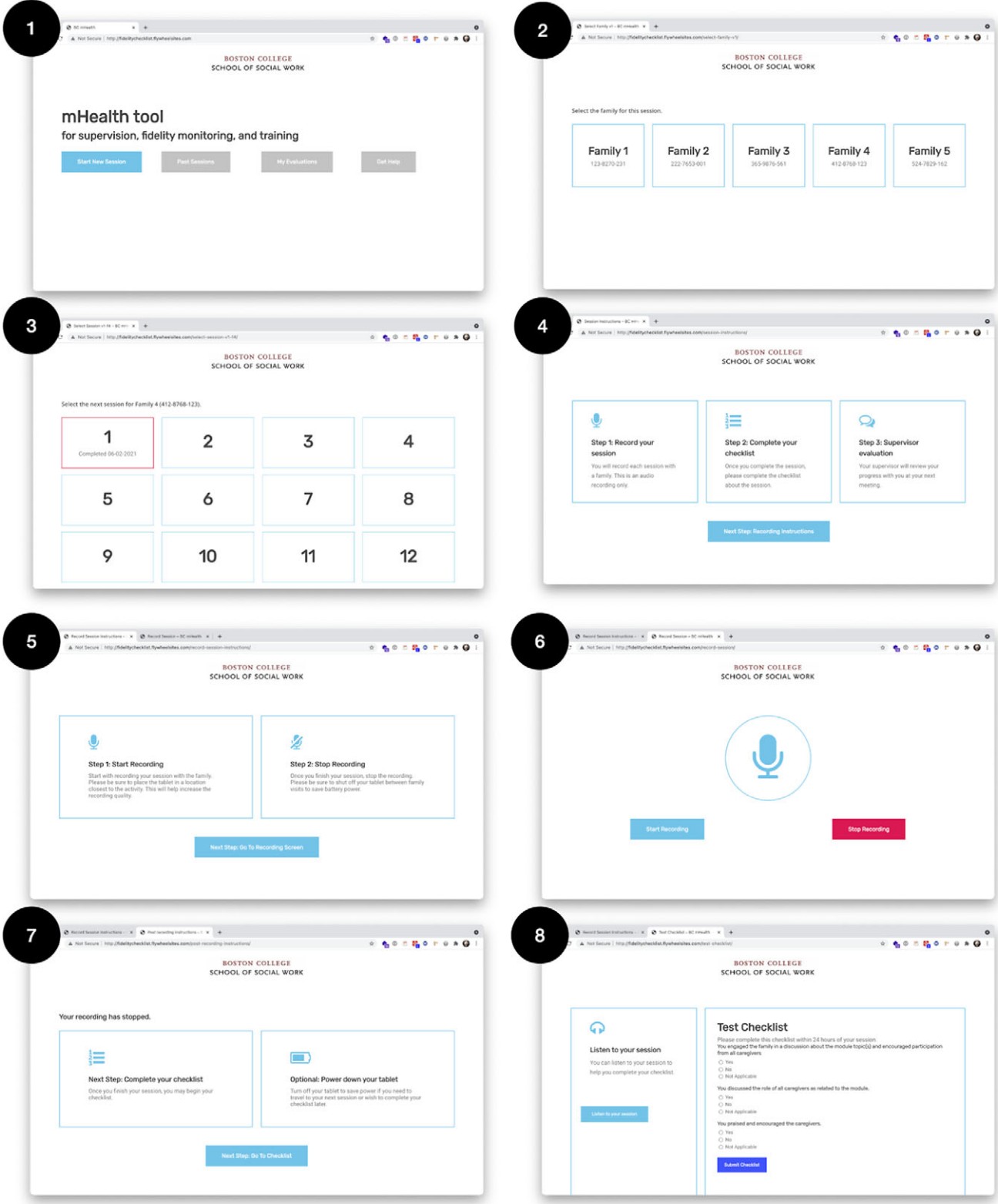

**Figure 2.** mHealth supervision tool prototype.

and goals and video tutorials that are accessible both online and offline. Provide ongoing technical assistance to CHWs and supervisors, either through in-person visits or remote support, could also improve usability in the future. The hybrid UCD methodology, while offering flexibility and reducing costs of international travel, presented challenges related to participant engagement and real-time interaction between participants and the research team. In-person engagement with the participants, coupled with remote

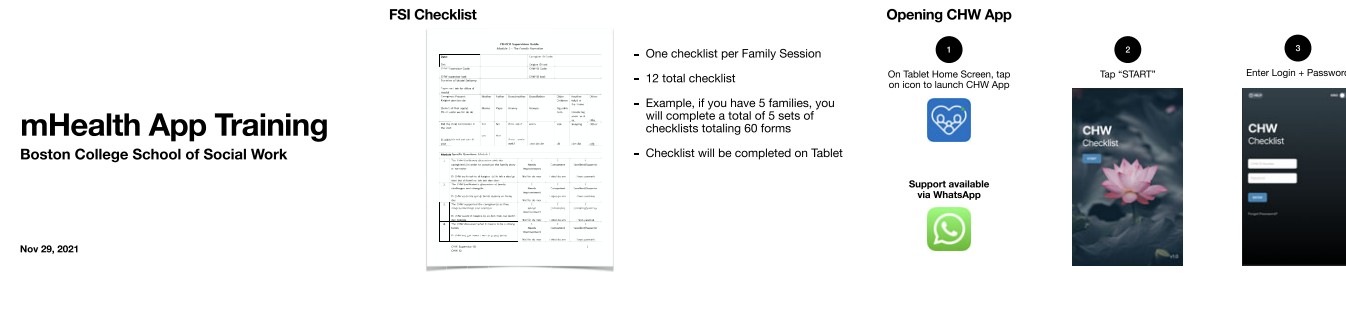

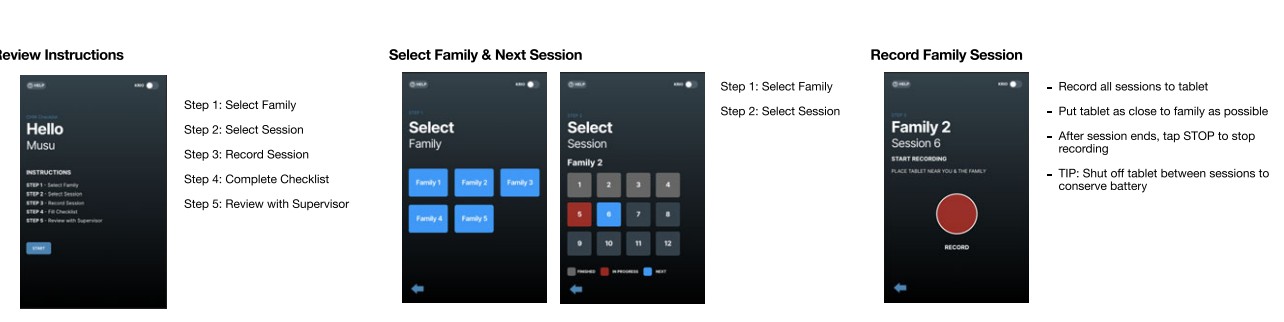

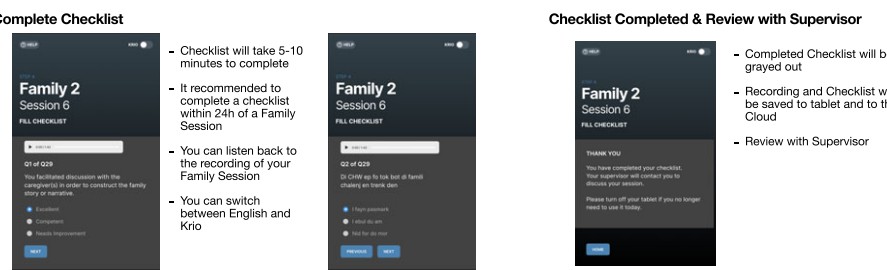

**Figure 3.** Final mHealth tool training for CHWs and supervisors.

teleconference sessions, allowed for direct interaction and leverage of the design team's expertise. However, the remote aspect of the approach might have limited the quality of interaction and rapport-building between the research team and participants, possibly affecting the depth and richness of the feedback obtained. Additionally, technical difficulties and inconsistent internet connectivity presented minor challenges for real-time communication and collaboration during the UCD process.

Despite these contextual challenges, the findings from this study underscore the potential for mobile tools to reduce barriers to EBI implementation and access in low-resource settings. The tool's flexible design and user-friendly interface make it adaptable to a variety of intervention contexts and content. Additionally, the UCD approach can be applied to the development of mHealth tools for other EBIs to help ensure that the tools are tailored to the needs and preferences of end-users. Findings also highlight the importance of understanding and then designing and developing mobile tools that incorporate features and functions to address contextual challenges (i.e., transportation issues, infrastructural and financial

constraints), which may limit the usability and ultimate scalability of mHealth tools in rural, resource-constrained settings. Future research could also consider strategies such as providing transportation allowances, integrating literacy support within the training or exploring alternative solutions for areas with limited connectivity to mitigate some of these challenges.

In conclusion, this study provides preliminary support for applying UCD methods to improve the acceptability and usability of mHealth tools to improve the supervision process and delivery quality of evidence-based behavioral interventions in low-resource settings. By centering the needs and preferences of end-users throughout the design process, it is possible to develop tools that are not only feasible and acceptable, but also highly usable and effective in LMICs and other resource-constrained settings. Findings highlight the importance of addressing contextual challenges, providing adequate training and support, and understanding the local technology infrastructure to maximize the benefits of mHealth tools in rural, resource-constrained contexts.

**Table 2.** Joint display of mHealth supervision tool system usability perceptions (N=8)

| Usability perception | Mean (pre-) | Mean (post-) | Change (pre–post) | Sig (p-value) | Illustrative quote(s) | Usability findings |
|---|---|---|---|---|---|---|
| I think that I would like to use this system frequently. | 4.5 | 4.75 | 0.25 | 0.334 | "It helps me a lot. I can now advise mothers better." (CHW 2) | Both quantitative and qualitative data suggest high user satisfaction and willingness to use the tool frequently, indicating it aligns with CHW needs for a practical and helpful resource to provide guidance and decision support in their daily work. |
| I found this system too complex. | 2.375 | 2.875 | 0.5 | 0.446 | "When it started initially it was not easy and it was a bit confusing. Whenever I listen to audio and compare it from the book… It helps me to learn a lot. It broadens my knowledge … I will look straight in the book and it helped me to learn very fast." (Supervisor 2) | While the quantitative data show a slight increase in perceived complexity, qualitative feedback emphasizes the overall ease of use and positive experience with the tool and resources provided to support participants. This suggests that the complexity might be manageable or even perceived as a feature offering more functionalities, as opposed to a burden. |
| I thought the system was easy to use. | 4.14 | 4.13 | −0.02 | 0.977 | "It's easy to use, I use it well." (CHW 2) "I like it because it directs me on what to do at home visits." (CHW 5) | Both quantitative and qualitative findings indicate high perceived ease of use of the mHealth tool, aligning with the user-centered principle of creating intuitive and user-friendly tools. |
| I think I would need help from a technical person to use this system. | 3.125 | 4.75 | 1.625 | 0.002 | "They should add the training." (CHW 3) "The training was not enough…they should repeat it." (CHW 6) | The significant increase in perceived need for technical support aligns with the qualitative feedback highlighting the need for more comprehensive and ongoing training, as well as access to technical assistance for troubleshooting. This suggests the tool may have some features that require additional guidance or clarification, especially during the initial adoption phase. |
| I found that the different functions and components of the system fit together well. | 4.25 | 4.5 | 0.25 | 0.334 | "They trained us how to use the tab because they gave us [a] manual that will tell you how the tab looks, how to use it, what and what you should do next, everything was there. If you forget how to use the tab you will just consult the book, which will give you a clear direction on how to use it." (CHW 3) | Convergence: The quantitative data indicate that users generally find the different components of the system to fit together well, which aligns with the user-centered design goal of creating a cohesive and integrated user experience. However, the lack of qualitative feedback from users on this aspect suggests it might not be a prominent factor in usability perceptions compared to other themes like ease of use or need for support. |
| I thought there was too much inconsistency in the system. | 2.125 | 1.875 | −0.25 | 0.448 | "The training helps me greatly…because the things that I supposed to use I was not having any difficulty in getting them, like the tablet as they have already trained us on how to use them on how to switch it on, what and what to do and the books they gave me also served as a help that makes me not to strain in doing the work." | While the qualitative data did not directly address the issue, the quantitative data suggest that CHWs perceived some inconsistency in the system. Qualitative feedback on navigation and feature challenges could be related to perceived inconsistency. Qualitative feedback also pointed to training and materials consistent with and supportive of system use. |
| I think most people would learn to use this system very quickly. | 3.5 | 3.75 | 0.25 | 0.678 | "At the training they directed us how to use it, it was difficult because we never used it before but as time goes we were able to use it and learn a lot from it." | While the quantitative data suggest a slight increase in the perception that the system is easy to learn, the qualitative data point to some CHWs finding it easy to learn and others needing more support. |
| I found this system hard to use. | 2 | 2.75 | 0.75 | 0.201 | "The only area that was having issues was the charging of the tablet." (CHW 2) "Well just like the raining season and the network was giving a hell of time" (Supervisor 2) | While the quantitative data suggest a slight increase in perceived difficulty of use from pre- to post-implementation, the qualitative evidence mostly emphasizes the tool's ease of use. However, the qualitative feedback on specific challenges (i.e., tablet charging and |

*(Continued)*

**Table 2.** (*Continued*)

| Usability perception | Mean (pre-) | Mean (post-) | Change (pre–post) | Sig (p-value) | Illustrative quote(s) | Usability findings |
|---|---|---|---|---|---|---|
| | | | | | | network connectivity) could explain the slight increase in perceived difficulty. |
| I felt confident using this system. | 4.5 | 4.5 | 0 | 1.000 | "One of the good things include: 1. The recordings, and 2. The app used was not giving us problems, with or without networks we were able to send our report and that was really good." (Supervisor 3) | Both data sources indicate that CHWs generally felt confident using the tool. However, the qualitative data reveal that some CHWs would benefit from additional training and support to further enhance their confidence, a difference not captured as clearly in the quantitative data. |

**Open peer review.** To view the open peer review materials for this article, please visit http://doi.org/10.1017/gmh.2025.38.

**Supplementary material.** The supplementary material for this article can be found at http://doi.org/10.1017/gmh.2025.38.

**Data availability statement.** Data used in this study are publicly available from ClinicalTrials.gov/NCT04481399.

**Author contribution.** CA and JP conducted mixed methods analyses for the study and conceptualized the presentation of key findings. CR and SB led UCD processes by facilitating codesign and development, leading user-testing workshops, developing training materials and documenting app development. MM and MF facilitated codesign processes in Sierra Leone, leading workshops in-country on behalf of the US-based research team. AD conceptualized the study, oversaw the analysis and provided a critical review of the manuscript. All authors approved the submitted version of the manuscript and agreed to be personally responsible for their own contributions.

**Competing interests.** The authors declare none.

**Funding information.** This work was supported by the National Institute of Mental Health (grant number R21 MH124071).

**Ethics statement.** The study protocol was approved by the Boston College Internal Review Board (Protocol #21.006.01) and the Sierra Leone Ethics and Scientific Review Committee. All participants provided oral informed consent before participating in the study.

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
