## [Reviewer Report]

Thank you for sharing this very interesting paper with me, I really enjoyed reading it, it is quite well-written and describes the development process in detail which makes it as easy and compelling read. There are a few minor points for consideration:

1. The tiile is User-centered design of an mHealth supervision tool to enhance the delivery of an evidence-based home visiting intervention (FSI-ECD+VP) for families in rural Sierra Leone

Considering the methodology and aim of the study, the title should reflect feasibility and acceptability in the title instead of simply saying enhancing

2. In introduction, lines 39-54 could include some details about how the innovative solutions would help with infrastructure and transportation related barriers

3. It would be good to know out of how many CHWs and Supervisors was this sample of N=8 and N= 4 selected and what was the sampling method?

4. Though there is some information given, would like to know more details about the kind of training and experience of the CHWs and Supervisors specific to the intervention being implemented

5. How was technical literacy of CHWs explored?

6. What was the mindmap and findings from the problem analysis?

7. Were the supervisors able to access the exact same information that was available to the CHWs on the app?

8. In analyze it is mentioned that there were 2 problem analysis FGDs while in data and procedures, 3 problem analysis FGDs are mentioned. Was there another FGD?

9. In Table 1, CHWs N is mentioned as 8 but in age columns the N adds up to 12, are the supervisors also included in it?

10. How will the discrepancy between qualitative and quantitative data be addressed as it doesn’t give any information about the inconsistencies in the system?

---

## [Reviewer Report]

This useful article helps to bring insight into how UCD is being leveraged in the field. A very useful read.

A few comments and suggestions for improving the manuscript are as follows:

1. The inclusion criteria do not state that the CHWs or supervisors had to have any experience with the Family Strengthening Intervention for Early Childhood Development plus Violence

Prevention; yet, throughout that paper, it seems that knowledge or experience in this intervention is critical to the testing, as it is part of the initial UCD goal. Would you clarify this in the article, either via the methods or throughout, whether knowledge in this area was needed, as well as whether focusing on this specific intervention may or may not limit the outcomes, such as translating it more broadly to supervision practices in this area?

2. Please add a general description of the proposed delivery and supervision process of FSI-ECD+VP, either in the intro or as part of the setting description in the methods.

3. The last paragraph of the intro includes a result/conclusion statement (line 40-47). This should be rephrased or removed.

4. The methods need more description from the design team - who they are, where they are, and how many they are. Please also be clearer on the schedule of the sessions that were remote vs in person, such as in a table. Also, itd be ideal to recognize if there was a single facilitator in the design team (the program manager?) of these sessions or multiple facilitators (which could be included when describing who this team is).

5. Who defined the end goal, was it the design team or the design team and the CHWs and supervisors?

6. Who is the program manager?

7. Curious why or whether current examples of similar platforms were used in any of the discovery processes (the first 2 stages) to leverage brainstorming or facilitate a platform that could be useful beyond this setting or intervention, as stated in the goal of the study?

8. Please include the list of questions used in the think aloud session, or at the least examples of the questions with an overall number or range, and detail whether these were asked in structure (all in one order) or flexible (skip around and not need to ask all) during the process.

9. Were any notes taken during observation? Also, was the observation remote or in person, was it a visual observation of how they used the tool by hand or via an app to capture swipes and clicks? Similarly, during the initial stage of discovery with FGDs, did any of the design team sit or observe the actual process of delivery or supervision (if given client permission) and/or visit the site to get sense of comms between CHWs and supervisors? Its not clear whether this type of observation happened.

10. line 28 in “Implement” says the product was finalised, though typically iterative changes should continue during implementation. Please clarify - is this a final product that will only change if more funding comes, or does the UI/UX allow iterative changes during implementation? I noticed further data collected via interviews and surveys for potential changes in the future, but is that really the case or are these all “nice to haves” if you can get to it?

11. Were the same CHWs and supervisors that designed it the ones that tested/implemented it / were there any blind CHWs or supervisors during testing?

12. authors write “brief training”. please elaborate via hours or days, remote or in-person. Also, the figure isn’t exactly clear, it seems like the training was only via an e-course-- was there a trainer that could give hands-on training with feedback? Finally, it describes the training as guiding “proper” use of the tool, but what does “proper” mean?

13. could you clarify if the phones belong to the CHWs and supervisors, or if they are supplied?

14. there is a lot of room to add to the discussion about the results including feedback on wanting to know more about “how to use it”. can the authors say more about how it was used, for instance: how much time does it take to fill out, do they have it out when they’re with clients, do they need permission, how does the permission go for the recording; how does this tool change or compare to the way supervision and fidelity was done before among CHWs; where are the recordings stored; do they need to listen to the whole recording to score and how does a score impact over multiple sessions; do CHWs use it independently as feedback for how to improve, or is that up to the supervisors discretion (what to improve, when), how often do CHWs and supervisors discuss results and is this the issue with the transportation? etc.

15. As a reader, the transportation issues are confusing. Is it that CHWs didn’t visit the home at first? Its hard to understand how that issue interferes with using the application, unless these results are also alongside testing the implementation of the home visits, the intervention itself, and consistent supervision between sessions?

16. Similar to point 15, incentives for who to use what? No incentive was described earlier about testing this app, so what is it, do they mean incentive to do recordings with the client or incentive to deliver the intervention?

17. It seems striking the language barrier wasn’t sorted prior, could the authors say more about this? for instance, was it because these are new users and not those included in the design, or is it related to the clients? Also, what part did the language interfere, useability of the app entirely (knowing where to click), the fidelity check list, all?

18. The discussion could be strengthened, for instance, why not discuss more on exactly what could be improved immediately, such as the training that was asked for by the participants? Similarly, it’s quite common that tech like this needs a helping hand, so why not discuss more on how the authors might answer to this in the future, or whether they will? Especially in terms of applicability beyond FSI.

---

## [Reviewer Report]

Thank you for this excellent paper demonstrating the important role of mHealth in reaching difficult to reach populations and the important role of CHW in such areas.

A follow up study with a larger N is recommended.

---

## [Editor Report]

There are quite a few concerns raised by one of the reviewers and the other reviewer as well. Hope you will be able to address all these concerns.